# Step-patterned survivorship curves: Mortality and loss of equilibrium responses to high temperature and food restriction in juvenile rainbow trout (*Oncorhynchus mykiss*)

**Jennifer L. Gosselin**ⓘ*, **James J. Anderson**

School of Aquatic and Fishery Sciences, University of Washington, Seattle, Washington, United States of America

* gosselin@uw.edu

**Data Availability Statement:** We have added our data to Dryad. https://doi.org/10.5061/dryad.47d7wm38v.

## Abstract

While survivorship curves typically exhibit smooth declines over time, step-patterned curves can occur with multiple stressors within a life stage. To explore this process, we examined the effects of heat (24˚C) and food restriction on juvenile rainbow trout (*Oncorhynchus mykiss* Walbaum) in challenge experiments. We observed step-patterned survivorship curves determined by mortality and loss of equilibrium (LOE) endpoints. To examine the cause of heterogeneity in the stress responses from early to late mortality and LOE, we measured indices of energetic reserves. The step transition in the survivorship curves, the peak mortality rates, and start of when individuals reached a critical energetic threshold (14% dry mass; 4.0 kJ·g$^{-1}$ energy) all occurred at around days 10–15 of the challenge. The coherence in these temporal patterns suggest heterogeneity in the cohort stress responses, in which an early subgroup died from heat stress and a late subgroup died from starvation. Thus, their endpoint sensitivities resulted in step-patterned survivorship curves. We discuss the implications of the study for understanding effects of multiple stressors on population heterogeneity and note the possible significance of stress response selection under climate change in which heat stress and food limitations occur in concert.

## Introduction

How multiple stressors affect organisms through their lives—and consequently their survivorship patterns—can depend on heterogeneity among individuals and the time scales of selective forces [1, 2]. Survivorship is often viewed as a smooth pattern acting uniformly over an entire lifespan or life stage [3, 4]. However, sequential stage-specific mortality can also be viewed in terms of step patterns in survivorship (e.g., a series of Type-III-patterned curves). Importantly, step-survival patterns across life stages may occur as individuals experience an intensification or relief from ecological stressors [5–7]. Even within a life stage, step-survival patterns can occur as individuals less able to tolerate a stressor are selectively eliminated. Identifying biological mechanisms underlying the pattern is important for understanding the force of selection

**Funding:** JLG and JJA received funding from Bonneville Power Administration, U.S. Department of Energy (grant number 00076910) https://www.bpa.gov/Pages/home.aspx JLG received funding from the Marsha Landolt and Robert Busch Endowed Fund, School of Aquatic and Fishery Sciences, University of Washington The funders did not play any role in the study design, data collection, analysis, decision to publish, or preparation of the manuscript.

**Competing interests:** The authors have declared that no competing interests exist.

across and within life stages. But again, at the heart of this selection process is heterogeneity among individuals and timing of stress responses.

Heterogeneity in a population's stress responses can be important in shaping its survivorship curve. Population trait variability can arise through adaptation and acclimatization [1, 8]. The expression of single or multiple genotypes across different environments and different cumulative experiences can result in different phenotypes; in turn, these phenotypes can shape the responses to stressors [2, 9, 10]. For example, when individual adult sockeye salmon swim upstream to their spawning grounds, their energetic reserves will be important as they endure increased metabolic rates under warm conditions [11]. Furthermore, individuals can also vary in their susceptibility to processes not primarily dominated by energetic reserves and metabolism [12], but more to acute processes such as heat shock that can affect the structural integrity of cells [13–15].

Given the projected increased frequency of extreme temperatures associated with global warming [16, 17], understanding how acute and chronic heat affect survival is important [18]. Many studies have separately examined short-term effects of heat stress and long-term effects of starvation, but few have looked at both together [19]. In general, responses from different stressors and their associated time scales can be particularly important in heterogeneous populations [13, 14]. Different biological mechanisms underlying the effects from heat include a systematic breakdown in thermoregulation and subcellular processes [20], and prolonged depletion of energetic reserves to critical levels [11, 21, 22]. Tracking the timing of both acute and chronic effects from lethal heat in juvenile fishes is feasible and informative in challenge experiments. These processes are also relevant to diverse systems, particularly ectotherms that experience more frequent, extreme, and chronically higher temperatures than those experienced historically [18, 23–25].

Patterns of survival can be produced in challenge studies of individuals subjected to stressors such as heat, absence of food resources, predation, pathogens, and toxins [26–28]. For this study, we focus on how thermal and caloric stressors can influence survival patterns. In particular, elevated water temperature has often been used as a stressor in challenge studies because the resulting mortality occurs over a time scale of days [29, 30], which is convenient for quantifying fish thermal tolerance [29, 31–33]. Caloric restriction has also been used in stress challenge studies and is useful for our purposes because it occurs over longer temporal scales of weeks to months [21, 34, 35]. Thus, the differing mechanisms of mortality and temporal scales of actions make the combination of thermal and caloric stressors desirable for studying the effects of population heterogeneity in forming step-patterned survivorship curves.

Observing the endpoint of LOE in place of mortality provides an arguably more ecologically relevant response. In context of experiments, surrogate endpoints are used to reduce the pain experienced by individuals for animal welfare, while still allowing investigation of mortality. LOE is often used as a surrogate endpoint in fishes because of their characteristic response to some stressors [36, 37]. As fish experience heat stress, the tail first drops, a phase at which they can still regain equilibrium; then they roll over (i.e. lose equilibrium) and cannot regain an upright position [36]. LOE has been used as a surrogate endpoint in many studies of critical thermal stress [29, 30, 37–39], but only averages or medians of time to LOE were reported. Given the multitude of processes possibly involved with heat, the timing of LOE may not relate directly to the timing of mortality. Thus, our study also investigates LOE as a possible alternative and ecologically meaningful endpoint for characterizing the response of fish to natural stressors.

By and large, mass population die-offs can occur, but more often only a portion of the population dies [10, 22, 40]. In the current study involving a model fish species, rainbow trout, we tested whether and how a die-off can occur in a heterogeneous population exposed to

challenge experiments at elevated water temperature and with food restriction. The study was primarily designed to characterize and compare the effects of acute and chronic stressors on survivorship patterns. A second goal was to identify the critical energetic threshold for the onset of mortality in juvenile fish. The third was to assess LOE as a surrogate endpoint in place of mortality. Altogether, this study provides a perspective on how heterogeneity in the population influences the type and the timing of responses to particular stressors.

## Materials and methods

### Fish, experiment, and data collection

We conducted challenge experiments at an elevated water temperature (24˚C) and in the absence of food resources at the University of Washington (Seattle, Washington) with juvenile rainbow trout (Nisqually Trout Farms, Inc., Olympia, Washington). The experiments were performed under protocol #3382–03 approved by the Institute of Animal Care and Use Committee (IACUC) at the University of Washington (UW). We designed the experiments with considerations in replacing live animals with alternative test subjects, minimizing the number of animals used, and using surrogate endpoints in place of mortality. Because the current study is on survivorship of fish, we could not replace the live animals with with cell culture or apply *in vitro* assays. We determined appropriate sample sizes for experimentation based on an earlier study [41]. We also minimized the number of animals used by not having a formal control group, but instead used a baseline group for comparison (more details below). With regards to applying a surrogate endpoint in our experiments, our study was in part aimed at investigating the appropriateness of loss of equilibrium in place of mortality. We were trained in decentralized animal care and certified by the UW Office of Animal Welfare prior to experimentation.

Prior to experimentation, the 200 rainbow trout tested were bulk weighed in groups of four to nine fish to minimize handling (average wet mass 6.8 g, see S1 Table). We placed the fish into a tank system with water re-circulating in parallel at a density of 25 individuals per 113.6-liter tank and acclimated them for 1.5 days at 9.5˚C using a water chiller. We monitored the water temperature with a logger (Maxim iButton® DS1922) every 10 minutes for the first 22 days. We monitored and maintained the water quality (temperature, pH, nitrite, ammonia, and dissolved oxygen) daily through the study. The light:dark cycle followed an 8:16 schedule.

We chose a target temperature for our challenge experiments that was below published upper incipient lethal temperatures and above that was above temperatures in which fish could maintain weight for several weeks. Upper incipient lethal temperatures have been observed at 25.6˚C [42], and at approximately 25.0˚C to 26.5˚C, depending on age and acclimation temperatures [43]. In a rainbow trout study, two stocks (Ennis and Winthrop hatchery strains, Montana, USA) and a population from a permanently heated stream in Yellowstone National Park (Wyoming, USA) had similar upper incipient lethal temperatures of 26.2˚C [44]. These were acclimated at temperatures as high as 24.5˚C and fed. We also chose a temperature that was high enough that rainbow trout would not survive the challenge experiments for multiple months. All fish within 1˚C of upper incipient lethal temperatures survived in short-term experiments [42]. At 24˚C and 60 days into an experiment with food rations, about 75% of rainbow trout can still survive and grow [45]. Furthermore, the inflection point of the survivorship curve was a little above 24˚C. Selecting this target temperature could help produce a wider range of responses in terms of time to mortality among heterogenous individuals. Thus, a temperature of 24˚C, which is also about 1˚C below the upper incipient lethal temperatures, was selected as an appropriate temperature for testing in our study.

We separated fish into different groups at the onset of the challenge experiment (i.e. once 24˚C was reached). We increased the water temperature to 24˚C at a rate of 0.2˚C·hr$^{-1}$ over

three days with submersible heaters. At the onset of the challenge, a random subsample of fish (n = 20) was euthanized with tricaine methanesulfonate (MS-222; 250 mg/L buffered to 7.0 with NaHCO$_3$) for the baseline group. Thereafter, 90 randomly-selected fish in four tanks remained in the "mortality group" and were sampled only at time of mortality. The "LOE group" also had 90 randomly-selected fish housed in another four tanks and were sampled when we observed that they had lost equilibrium and could not regain an upright position. These fish were euthanized with MS-222 at time of sampling. We determined a sample size of 90 from goodness-of-fit tests of related studies [41, 46] during the development of our animal care protocol. A formal control group tested at the temperature at which they were raised, and replicates for each treatment group would have helped strengthen our experimental design; but we chose to minimize the number of individuals tested for animal ethics and welfare. Thus, care is needed in interpreting results without formal controls and replicates.

We observed the fish for mortality and LOE at least 3 times a day at approximately 09:00, 16:00, and 22:00. We made observations at finer temporal resolution opportunistically. But given the total duration of the challenge experiment, observations at these finer temporal scales were not necessary in our analyses. If any fish showed visible signs of disease, we euthanized and removed them from the experiment. This only occurred to one fish in the mortality group. During observations, a number of fish in the LOE group died before any signs of equilibrium loss were detected, and were sampled at their time of mortality. This likely occurred because of different processes controlling LOE and mortality, which we later discuss. As the experiment continued, we consolidated fish into fewer tanks to maintain approximately the same fish densities.

We recorded the wet mass of dead and euthanized fish individually and then froze them for subsequent processing to measure percent dry mass and energy density. We dried the fish at 60˚C and weighed them individually for dry mass. For approximately every other fish sampled during the challenge experiment, we quantified the energy density with a bomb calorimeter (SemiMicro Bomb Calorimeter 1425, Parr Instrument Company ®).

## Analysis

Because heat and the absence of food can produce mortality or LOE, but presumably at different rates, we hypothesized that a heterogeneous population would exhibit a heterogeneous response to the two stressors. This would imply that the faster acting stressor dominates in early challenge mortalities and the slower acting stressor dominates in late challenge mortalities. We characterized their respective dominance by assuming the step pattern was produced by two superimposed survivorship curves: one characterizing the mortality or LOE from heat stress and the other from starvation. Following [13], we assumed a fraction $p$ of each subgroup experienced mortality or LOE from heat stress and the remaining fraction 1– $p$ experienced mortality or LOE from starvation. Characterizing the survival pattern from each subgroup by a Gompertz survival function [47] the observed survival is defined

$$S(T) = pe^{-a_1(1-e^{b_1 T})} + (1-p)e^{-a_2(1-e^{b_2 T})} \tag{1}$$

where $S$ is survival (or proportion alive) at challenge time $T$, the parameters $a_1$ and $a_2$ are asymptotes and $b_1$ and $b_2$ characterize the mortality (or endpoint) rate increase with age. We estimated the model parameters with non-linear least squares in R© 2019 The R Foundation for Statistical Computing (version 3.6) with the function nls2 from R package nls2 version 0.2. The ranges of parameters tested were: $0 \leq p \leq 0.4$, $-0.01 \leq a_1 \leq 0$, $0 \leq b_1 \leq 0.6$, $-0.03 \leq a_2 \leq 0$, $0 \leq b_2 \leq 0.2$. We also determined the mortality and LOE endpoint rates ($dS/dT$) to visually compare to their corresponding survivorship curves.

To further test whether heat stress dominated the early mortality and LOE endpoints and starvation dominated the later ones, we examined the temporal pattern of energetic reserves to see if they would change over the course of the experiment. Energetic reserves are predicted to linearly decline until a critical threshold ($y_{crit}$) is reached, as observed by other studies [35, 48]. Thus, mortality and LOE endpoints from starvation should be associated with an approximately fixed critical value of energetic reserves, independent of when mortality occurs. In contrast, energetic reserves are not likely to directly influence mortality and LOE endpoints from heat stress such that no critical threshold is expected to become apparent. Rather, a declining linear relationship may arise because individuals are depleting their energetic reserves the longer they are in the challenge. Furthermore, the energetic reserves of fish dying from heat stress should be greater than the critical energetic threshold.

To determine the critical time when the dominant mortality process switched from heat stress to starvation, we regressed the energetic reserves index of percent dry mass ($M$) at mortality against their time at mortality. We tested a linear regression that incrementally left-truncated the regression interval. In this way, the first regression included a regression between $M$ and $T$ using all fish. The second regression left-truncated the earliest data point, the third regression included the next data point in the left truncation and so on until the regression contained only the last three data points. We reasoned that after sufficient left truncations, that time point onwards would represent mortalities from starvation. The corresponding slope of the regression would converge to zero and represent mortality occurring at a constant $M$. Once a left-truncated data set revealed a slope parameter that was not significantly different from zero at $\alpha = 0.05$, we designated a critical time ($T_{crit}$) that corresponds to the transition from heat-stress-dominated to starvation-dominated mortality as the time when the slope's 95% confidence interval no longer intersected zero.

We estimated the critical energetic threshold utilizing only fish in the left-truncated data set. We determined the threshold in units of percent dry mass which was converted to energy density (kJ·g$^{-1}$) by applying a transformation utilizing the log-linear relationship we determined between our energy density and percent dry mass data.

We analyzed the mortality and LOE groups separately for fitting survivorship curves, estimating the onset of $T_{crit}$, and determining critical energetic thresholds. In comparison to the mortality group, we expected the LOE group to show a comparable or higher endpoint rate, an earlier $T_{crit}$, and a higher critical energetic threshold.

## Results

We observed a step pattern in the survivorship curves of juvenile rainbow trout introduced to a heat challenge experiment in the mortality group (Fig 1A) and the LOE group (Fig 1B). The timing of the step patterns in both curves were comparable to when an increase in their mortality and LOE endpoint rates occurred: early in the challenge at around 10–15 d (Fig 1C and 1D). In both survivorship curves, heterogeneity within each group was characterized by the $p$ proportion of the mortality group and of the LOE group that experienced an endpoint in the early period (Table 1). The early subgroups had higher endpoint rates compared to the late subgroups (respectively, $b_1$ and $b_2$ in Table 1). Similarly, the early LOE endpoint rates (Fig 1D) were generally greater than the early mortality rates (Fig 1C). Thus, a larger $p$ was estimated in the LOE group than the mortality group. This pattern suggests that LOE serves as an alternative endpoint that identifies a larger portion of the population under stress and forthcoming mortality. We suggest the LOE endpoint is more ecologically relevant endpoint because any fish experiencing LOE is highly susceptible to predation.

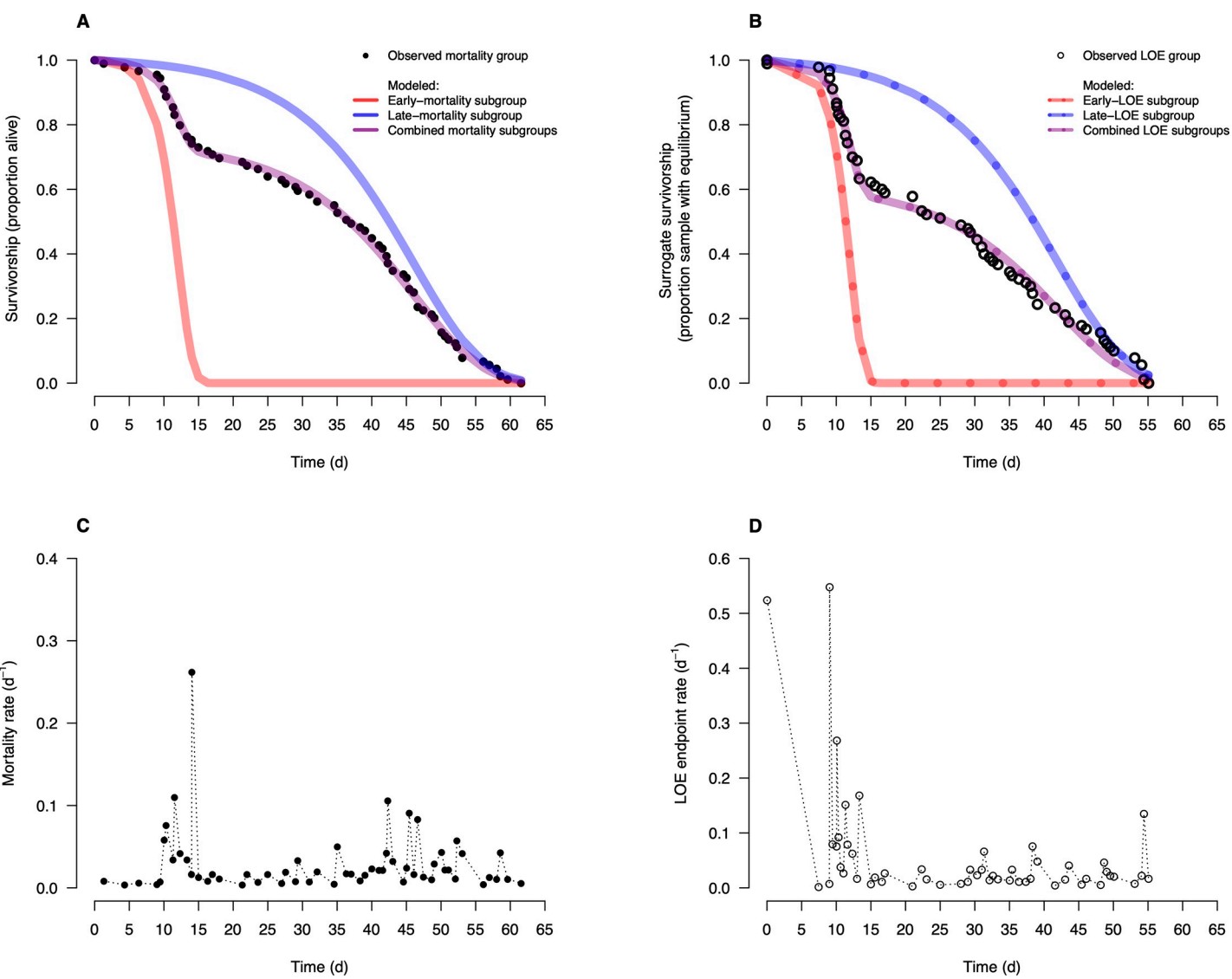

**Fig 1.** Survivorship curves of rainbow trout and their mortality rate $\left(-\frac{\log_e(N_t/N_{t-1})}{T_t-T_{t-1}}\right)$ at times of $t$ observations in a challenge experiment with stressors of heat and absence of food. (A) Survivorship based on mortality endpoint. (B) Surrogate survivorship based on LOE endpoint. (C) Mortality rate. (D) LOE endpoint rate. Modeled survivorship (purple) is a mixture model of two Gompertz functions (Eq. 1), respectively for early (red) and late (blue) subgroups.

The percent dry mass of fish sampled at mortality and LOE endpoints generally showed a steeper decline early in the challenge compared to mid-late in the challenge (Fig 2A and 2B). These results support the hypothesis that fish of one phenotype did not die primarily from

**Table 1. Survivorship model parameters of Eq 1 for mortality and LOE groups estimated by non-linear least squares.**

| Parameter | Mortality group | LOE group |
| --- | --- | --- |
| $p$ | 0.263 | 0.394 |
| $a_1$ | −0.003 | −0.002 |
| $b_1$ | 0.480 | 0.540 |
| $a_2$ | −0.010 | −0.015 |
| $b_2$ | 0.100 | 0.100 |

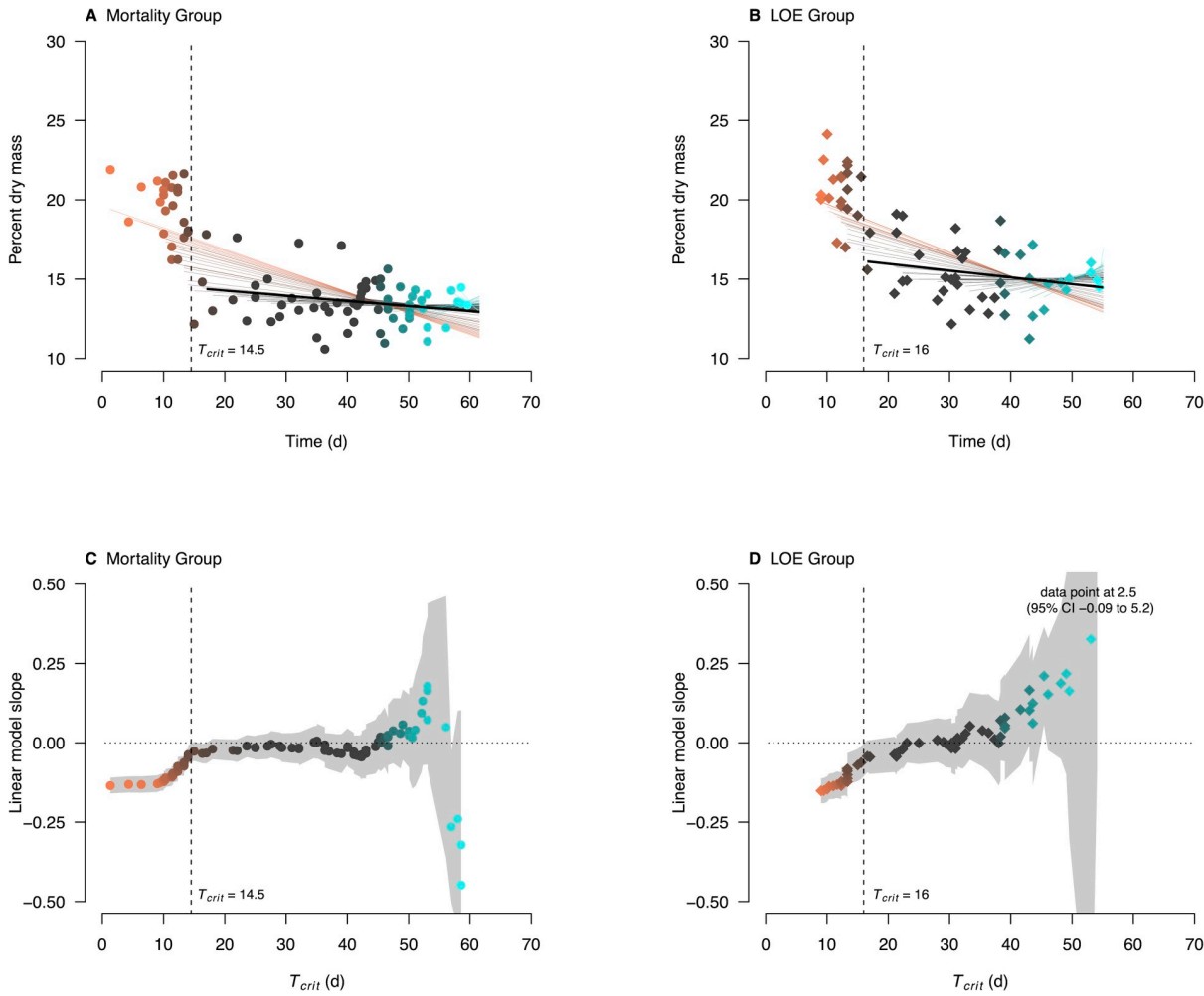

**Fig 2. Energetic reserves measured as percent dry mass, and associated linear model slope parameters over time of challenge experiment.**
Energetic reserves at time of (A) mortality and (B) LOE. Slope parameters of progressively left-truncated data sets for the (C) mortality group and (D) LOE group. Points represent slope parameters and gray shading represents the points' respective 95% confidence intervals. Data points and linear model slopes are color-coded by the left-most point analyzed. Vertical dashed lines represent $T_{crit}$ when the 95% CI of the slope parameter begins to overlap a slope of zero. Solid black lines represent the linear model with the associated $T_{crit}$.

starvation early in the challenge, but primarily from processes associated with acute heat stress. The timing of $T_{crit}$ demarcating the switch from heat stress to starvation (Fig 2C and 2D; Table 2) was comparable to when the mortality and LOE endpoint rates increased at around 10–15 d (Fig 1C and 1D).

The $y_{crit}$ critical energetic thresholds were about 14% and 15% dry mass for the mortality and LOE groups, respectively (Table 2). These thresholds were lower than the mean percent dry mass of the baseline group (23.0% ± SD 2.0%, n = 20) and of the early subgroups (19.0% ± SD 2.4%, n = 25 for mortality group; 20.6% ± SD 1.8%, n = 19 for LOE group). Within each of the early and late subgroups, mean percent dry mass was higher for the LOE group than the mortality group. This is expected because LOE occurred before mortality (S1 Fig), and the fish had not depleted as much energy at this earlier endpoint than at mortality. Furthermore, the larger coefficient of variation of $y_{crit}$ in the LOE group (0.122) compared to the mortality group (0.108) may indicate that factors other than energetic reserves controlled equilibrium.

**Table 2. Critical times ($T_{crit}$) and critical energetic thresholds ($y_{crit}$).** These were determined from linear regressions between percent dry mass and time of mortality or LOE endpoint that statistically showed a zero slope ($n_{crit}$ sample size; thick black lines in Fig 2). $T_{crit}$ represents the time when heat stress switches to starvation stress, and $y_{crit}$, represents the critical energetic thresholds in percent dry mass estimated as intercepts only. Values in parentheses represent 95% confidence intervals. N is the total number of individuals tested in the challenge experiment and measured for percent dry mass.

| | Mortality group | LOE group |
|---|---|---|
| **intercept** | 14.9 (13.5, 16.3) | 16.8 (14.9, 18.7) |
| **slope** | -0.032 (-0.065, 0.000) | -0.043 (-0.093, 0.008) |
| $T_{crit}$ | 14.5 | 16.0 |
| $y_{crit}$ | 13.6 (10.7, 16.4) | 15.3 (11.6, 18.9) |
| $n_{crit}$ | 64 | 45 |
| **N** | 89 | 64 |

Energy density was strongly correlated with percent dry mass (Fig 3, $r$ = 0.915). Following the log-linear relationship we determined, the critical thresholds of 13.6% and 15.3% dry mass were equivalent to energy densities of 4.0 kJ·g$^{-1}$ and 4.3 kJ·g$^{-1}$, respectively.

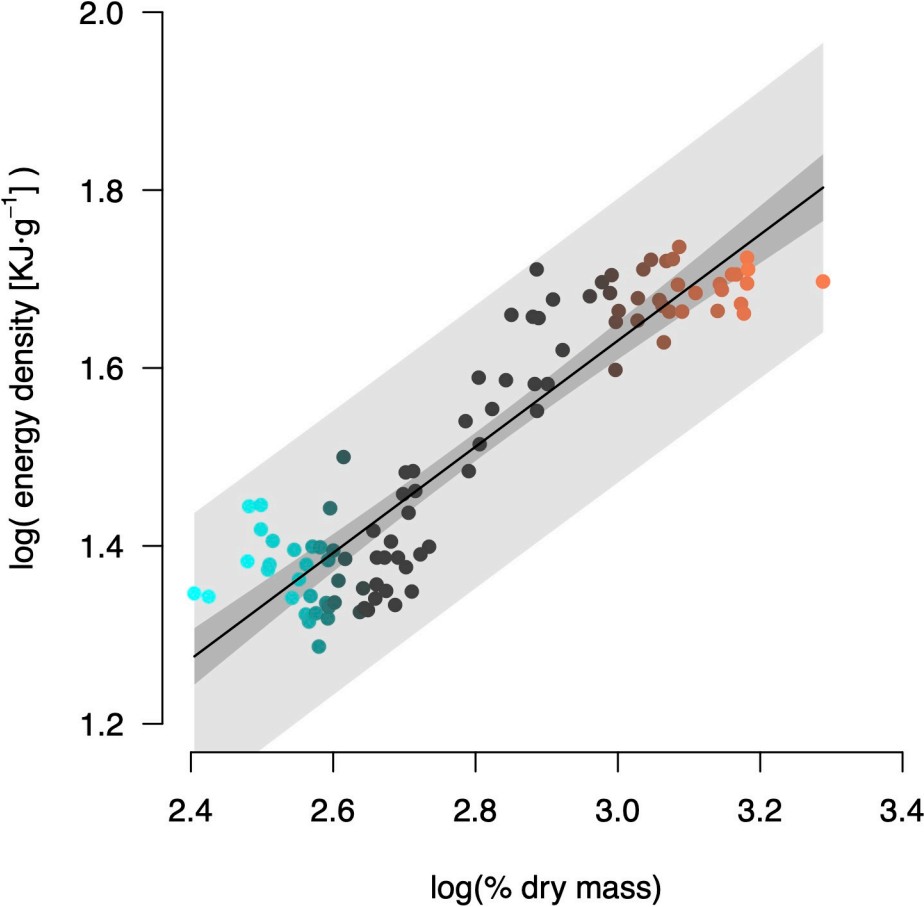

**Fig 3. Log-linear relationship between energy density ($E$) and percent dry mass ($M$): log($E$) = −0.159 + 0.597 × log ($M$); $r$ = 0.915; $p <$ 0.001; n = 99.**

## Discussion

### Population survival bottlenecks and individual heterogeneity

A population can undergo a number of bottlenecks with sharp declines of survival as individuals progress through their life cycle stages [5, 6]. These bottlenecks occur particularly during episodes of extreme conditions, migration to a new habitat, and ontogenetic shifts [18, 22, 40]. Anadromous fishes can experience high morality when they encounter new habitats, particularly across the freshwater, estuarine and marine systems [49]. Furthermore, many studies across various taxa, including invertebrates and vertebrates, have focused on the larval and juvenile life stages because of their importance to recruitment [50, 51]. For example, higher spring and summer temperatures can affect the physiology and survival of larvae and juveniles [18, 52], and higher winter temperatures can quicken depletion of energetic reserves [22]. Acute heat stress and energetic depletions are common stressors that many taxa experience through their life cycles and result in significant periods of mortality.

Population heterogeneity can result in survivorship patterns that reflect bottlenecks [13, 14]. A step in the survivorship pattern can result from two phenotypes exposed to a common stressor. In our study, the curve of each subgroup associated with the different time scales of the selection processes: heat stress within a few weeks in some individuals, and depletion of energetic reserves within a couple months in the remaining individuals. The timing of acute and chronic responses in our current study can be viewed akin to summer temperatures that reach the upper critical thermal limit and energetic reserves that reach critically low levels in summer or winter.

We inferred differences in heterogeneity among individuals from their survivorship patterns and energetic reserves because we did not measure specific indices of heterogeneity among individuals, nor did we have detailed information on the genetics or stocks of the fish from the distributor. We note that rainbow trout can grow at higher temperatures, and thus over a wider range of temperatures, compared to other trout species like westslope cutthroat trout (Bear et al. 2007). Given their wide range of temperature tolerances, this may allow greater heterogeneity among individuals and a wider range of responses. Had we been able to track also each individual's level of energetic reserves from beginning to end of the challenge experiment, we would have been able to better infer whether the times to energetic depletion were due to their initial states and/or their rate of energetic depletions [53]. Below, we discuss possible underlying differences in their heterogeneity. Because we measured indices of energetic reserves, we first discuss starvation stress and critical energetic thresholds. We then consider possible processes involved in heat stress.

### Starvation stress and critical energetic threshold

Our hypothesis that individuals dying late in the challenge predominantly due to starvation and caused by a lack of food resources has strong support in the literature. The energy density critical thresholds of 4.0 kJ·g$^{-1}$ at mortality and 4.3 kJ·g$^{-1}$ at LOE are similar to 4 MJ·kg$^{-1}$ observed in migrating adult sockeye salmon [11, 54, 55]. Although, lower values (2.9 kJ·g$^{-1}$) have been observed in adult sockeye salmon at time of mortality [56]. Additionally, young-of-the-year walleye Pollock died when they reached 15% dry mass on average [57], which is comparable to the 14% dry mass threshold we observed at mortality and 15% dry mass at LOE in our study.

Survival bottlenecks depend on numerous factors related to energetic reserves such as temperature, food resources [58], and predation risk [59]. Furthermore, energetic reserves can be related to how individuals are heterogenous in their abilities to cope with stress, metabolic

rates [21, 60], levels of hormesis [14], phenotypic and genetic pre-disposition of behavioral traits [61], and energy allocation and storage between ecotypes [62–64]. The interactions among some of these factors add to ecosystem complexity with ultimate effects on survival [e.g., 65].

## Heat stress

Lethal heat stress has been hypothesized to be linked to systematic breakdowns spanning sub-cellular to whole-organism functions comprising many different kinds of biological indices [20, 66, 67]. Thus, heat stress is more complex than starvation stress, which overall has a common index of energetic reserves. To gain insight on what kinds of heat stress indices could be measured for population heterogeneity and critical thresholds in future studies, we summarize a number of ways heat stress can affect biological processes at the subcellular level and in the central nervous system below.

A link between an organism's temperature tolerance and the temperature limit of 70 kDa heat shock proteins (hsp70) has been determined in several studies [68–70]. Hsp70 repair denatured proteins and translocate permanently damaged ones to lysosomes and proteosomes for breakdown and removal [71–73]. Differences in the magnitude of induction among isoforms and the levels at which they are activated may explain differences in temperature tolerances among populations [74]. Differences within species adapted to different thermal environments can vary more widely than between species adapted to the same environment [75]. In a study of salmon in the wild, there was large variation in hsp70 levels on the spawning grounds [55]. Interestingly, individuals with higher hsp70 levels, that were induced by means of any kind, had lower energetic content than the rest of the individuals. The high hsp70 levels may have allowed these fish to tolerate environmental conditions and survive longer, until they depleted their energetic reserves. Overall, proportions of different phenotypes related to heat shock protein concentrations could explain the different susceptibilities to lethal heat-related stress.

Also at the subcellular level, oxidative stress could influence the thresholds of lethal heat stress [76, 77]. Reactive oxygen species (ROS) build up in cells as a result of aerobic metabolism and from environmental exposure[78]. In turn, an excess of ROS can damage deoxyribonucleic acid, proteins, and lipids and result in cell death [79].

Lethal heat stress can also occur with a thermoregulation breakdown stemming from the central nervous system [20, 37, 78, 80, 81]. A breakdown in the brain is thought to occur before a breakdown of cells in the rest of the body. This has been observed in eels placed in a swim chamber where the anterior and posterior halves of their bodies experienced different temperatures [82]. Generally, the hypothalamus receives afferent sensory information about temperature and then can initiate an efferent sympathetic response such as vasoconstriction and cellular metabolism [20]. Thus, thermoregulation, is in part, a neural process that helps maintain metabolism and stability in reaction rates within an organism in response to temperature change [78, 83].

In general, heat stress can involve a loss in the precision of biochemical regulation and overall system destabilization. Heat stress can negatively affect multiple other processes, including: cytoskeleton dynamics for cell motility, division, signaling and muscle contraction [84]; limited oxygen transport, elicitation of anaerobic metabolism, and consequently reduced aerobic scope [67]; and diseases [85]. Essentially, heat stress likely results in a conflict at the cellular and organ levels to maintain regular activities and to produce the ability to resist extreme temperatures [86, as cited in 73].

In our study, the fish that survived the early period may have had more genes activated or effectiveness in gene expression of proteins to counteract heat stress and maintain cellular integrity and homeostasis. These individuals could have already been exposed to other stressors, produced stress proteins, and were thus readily acclimated to heat stress, a process termed hormesis [14]. Although determining the exact process by which the individuals died in the early period of the challenge was beyond the scope of the current study, the individuals likely died from heat stress and not exclusively from a lack of energetic reserves.

## LOE endpoint in place of mortality

In contrast to mortality, LOE may be a more ecologically relevant endpoint because compromised swimming ability would increase susceptibility to predation. Our study supports LOE as a surrogate endpoint to mortality given their similar survivorship curves, timing of $T_{crit}$, and energetic thresholds. These similarities occurred even without formal replicates for each treatment group for animal ethics and welfare reasons. Still, we note caution in interpretation of results. An average across replicates would provide a more objective estimate of times between mortality and LOE. In light of the importance and utility of LOE as an endpoint, further research into its underlying mechanisms is warranted. Equilibrium is regulated by the central nervous system, but the control originates from the cerebellum [87] in conjunction with the vestibular system [88]. Abnormal sensations from a compromised vestibular system can result in forced rolling and circulus movements towards the affected side [88], but may not affect mortality directly. Many aspects of the vestibular system and its functions have not been sufficiently studied in fishes [89], such as the effects of heat and starvation. Overall, the patterns in survivorship and energetic reserves we observed were comparable between the mortality and LOE groups, thus supporting LOE as a suitable endpoint for ecological research of fishes.

## Conclusion

Overall, migratory species can encounter survival bottlenecks linked to factors such as energetic, nutritional, and disease-risk status of individuals [90]. In our study, there were likely two main phenotypes: one mostly incapable of acclimating to the heat stressor and dying before starvation occurred, and one mostly capable of acclimating to the heat stressor but then dying from starvation. Whether this is only a phenotypic distinction generated from different experiences, or whether the differences are rooted in their genes, remains unclear in the current study. This difference would be important to resolve to better understand their acclimatization capabilities and for species conservation in a changing climate. Furthermore, individuals that have evolved multiple and independent physiological mechanisms to cope with stressors could increase their chances of survival [91]. Our study has shown that step patterns in survival can be indicative of phenotypic variation among individuals and selection processes occurring at different time scales. Challenge studies can thus provide an index of the proportion of the population susceptible to stressors.

With the advent of increased intensity, frequency, and duration of heat waves above critical temperatures [92], more bottlenecks and step patterns in survivorship are likely to occur. Heat stress and tolerance relates to many different systems, including amphibians, corals, tidepool-dwelling organisms, birds, insects and trees [18, 23–25, 93]. It is thus important to understand heterogeneity in responses to competing stressors. Our study demonstrated that the pattern of mortality resulted from heterogeneity in the rates of response to heat and lack of food resources. In general, such heterogeneity is expected to involve phenotypic plasticity and genetics [94] within the population and clarifying its proximate causes will enhance our understanding of how organisms respond to temperature-related stressors.

## Supporting information

**S1 Table. Mass of rainbow trout by bulk group of individuals, and average mass by bulk group, tank, treatment group and all rainbow trout tested.** Standard deviation among individual fish are not available because fish were bulk weighed to minimize handling. The 10 individuals sampled from each of the treatment groups for the baseline group are included in these observed masses.
(DOCX)

**S1 Fig. Predicted difference in time to mortality ($T_{Mortality}$) and time to loss of equilibrium ($T_{LOE}$) associated with the fitted survivorship curves (Fig 1A and 1B in the main paper).**
(TIFF)

## Acknowledgments

We are grateful to David Beauchamp and his research lab group for loaning us their bomb calorimeter, and to Kerry Naish and her research lab group for loaning us their aquarium system at the University of Washington. We thank Jon Wittouck for loaning us laboratory equipment and for help with experimental setup of the fish aquarium system. We also thank David Beauchamp and Nick Beer for their comments on earlier drafts that helped improve our manuscript.

## Author Contributions

**Conceptualization:** Jennifer L. Gosselin, James J. Anderson.

**Data curation:** Jennifer L. Gosselin.

**Formal analysis:** Jennifer L. Gosselin.

**Funding acquisition:** James J. Anderson.

**Investigation:** Jennifer L. Gosselin.

**Methodology:** Jennifer L. Gosselin, James J. Anderson.

**Supervision:** James J. Anderson.

**Writing – original draft:** Jennifer L. Gosselin, James J. Anderson.

**Writing – review & editing:** Jennifer L. Gosselin, James J. Anderson.

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
