## [Decision Letter · Decision Letter 0]

27 Dec 2019

PONE-D-19-30081

Step-patterned survivorship curves: mortality and loss of equilibrium responses to high temperature and food restriction in juvenile rainbow trout (*Oncorhynchus mykiss*)**

PLOS ONE

Dear Dr. Gosselin,

Thank you for submitting your manuscript to PLOS ONE. After careful consideration, we feel that it has merit but does not fully meet PLOS ONE’s publication criteria as it currently stands. Therefore, we invite you to submit a revised version of the manuscript that addresses the points raised during the review process.

We would appreciate receiving your revised manuscript by Feb 10 2020 11:59PM. To enhance the reproducibility of your results, we recommend that if applicable you deposit your laboratory protocols in protocols.io, where a protocol can be assigned its own identifier (DOI) such that it can be cited independently in the future. For instructions see: http://journals.plos.org/plosone/s/submission-guidelines#loc-laboratory-protocols

*Please include the following items when submitting your revised manuscript:*

*A rebuttal letter that responds to each point raised by the academic editor and reviewer(s). This letter should be uploaded as separate file and labeled 'Response to Reviewers'.**A marked-up copy of your manuscript that highlights changes made to the original version. This file should be uploaded as separate file and labeled 'Revised Manuscript with Track Changes'.**An unmarked version of your revised paper without tracked changes. This file should be uploaded as separate file and labeled 'Manuscript'.*

*Please note while forming your response, if your article is accepted, you may have the opportunity to make the peer review history publicly available. The record will include editor decision letters (with reviews) and your responses to reviewer comments. If eligible, we will contact you to opt in or out.*

*We look forward to receiving your revised manuscript.*

Kind regards,

Madison Powell, PhD

Academic Editor

PLOS ONE

Additional Editor Comments:

*Two reviews provide modest but substantial edits which would greatly enhance the manuscript. Particularly, Reviewer 1 suggests several recent publications of interest to this subject area and at least an acknowledgement of these other studies needs to be included within the revised text. As pointed out in the reviews, the manuscript is generally well written and the study is easy to follow. The authors should also place a bit more emphasis on the relevance of their findings since temperature tolerance studies are being viewed across many different disciplines. Reviewer 2 provides a valuable editorial point regarding the endpoint of the experiment. It is exceedingly important to justify this and why it made a difference in the data generated for this manuscript.*

**2. **We note that you have stated that you will provide repository information for your data at acceptance. Should your manuscript be accepted for publication, we will hold it until you provide the relevant accession numbers or DOIs necessary to access your data. If you wish to make changes to your Data Availability statement, please describe these changes in your cover letter and we will update your Data Availability statement to reflect the information you provide.

**3. **Your ethics statement must appear in the Methods section of your manuscript. If your ethics statement is written in any section besides the Methods, please move it to the Methods section and delete it from any other section. Please also ensure that your ethics statement is included in your manuscript, as the ethics section of your online submission will not be published alongside your manuscript.

**

Reviewers' comments:

Reviewer's Responses to Questions

**Comments to the Author**

1. Is the manuscript technically sound, and do the data support the conclusions?

Reviewer #1: No

*Reviewer #2: Yes*

*2. Has the statistical analysis been performed appropriately and rigorously? *

Reviewer #1: Yes

*Reviewer #2: I Don't Know*

*3. Have the authors made all data underlying the findings in their manuscript fully available?*

Reviewer #1: Yes

*Reviewer #2: Yes*

*4. Is the manuscript presented in an intelligible fashion and written in standard English?*

Reviewer #1: Yes

*Reviewer #2: Yes*

*5. Review Comments to the Author*

Reviewer #1: This study investigated the survivorship of rainbow trout under heat stress and food restriction. By comparing mortality and LOE, the authors found a step-wised survivorship curve, with the acute response to be associated with heat stress and the chronic response to be associated with limited energy reserve. This manuscript is well present and easy to follow. However, the authors may want to provide some explanation to clarify some aspects.

It looks like the authors could provide more reviews of existing literature on studied topics. The upper lethal temperatures for rainbow trout are right around 24C, according to previous studies. (25.68C: Hokanson et al. 1977; 26.28C: Kaya 1978; Bear & MCMAHON 2007 and maybe more). That will provide background for readers about thermal requirement and tolerance of rainbow trout. Some of these studies also found acute mortality when acclimate fish at 24C, some fish were able to survive longer. Some of these papers also studied growth rate at high temperatures, which could be related to the chronic mortality due to limited energy reserves. Some thoughts on individual variability in the time of reaching the critical energetic threshold could be a nice addon.

While LOE is ecological relevant, its reduced mobility of escaping from lethal condition often leads to mortality. Under thermal stress, LOE and mortality is very close, as seen in many critical thermal maximum studies. I wonder how mortality were controlled in the LOE tanks (see some of the minor comments). Other than that, I have some minor suggestions:

Minor comments.

L47, I am not sure why acute heat shock is a “non-energetic process”. Also, some fish examples will be more appropriate than C-elegans here.

L74, Endpoints of LOE and mortality are often very close, especially when fish are under heat stress, e.g. critical thermal maximum. Did you expect to see a prolonged response of LOE here? In the later M&M section, you mentioned checking fish 3 times a day at 09:00, 16:00, and 22:00. LOE for several hours seem like a long time. How many fish from the LOE tanks died?

L94, as the author mentioned “a heterogeneous population” before. I am curious how closely related are these hatchery fish, assuming there are some degrees of inbreeding.

L97-100, these are probably unnecessary details, which are often seen in the AUP protocol.

L115 Some explanation of the choice of 24C. Some information/discussion of rainbow trout lethal temperatures will be useful (e.g. Bear et al., 2007). 24C seems like good choice as it is a lethal temperature for rainbow trout, but some fish could get acclimated and survive for a longer time.

L123 Same comment as L74.

L125 Same comment as L74.

L135 parenthesis are often used as explanatory.

L191 I am not fully understanding the “the LOE group that experienced early mortality”.

L193-194 Since LOE occurrs before mortality events and they are related. Are you able to calculate the closeness of the timing between mortality and LOE?

L204, this is more like a statement of the result, rather than a hypothesis. It also occurred in other sections.

L234 “Table 1” should be “Table 2”.

L238 Maybe the reason can be given for why “mean percent dry mass was higher for the LOE group than the mortality group”?

L326 Add year of reference

L326 “spontaneous mortality” needs a definition

L358-359 repetition of the statement made before (L355-L356)

*Reviewer #2: I was asked to review this paper looking specifically at the area of animal ethics and welfare. While I have some strong concerns regarding choosing death from either starvation or heat shock as an endpoint, my concerns were at least partially mitigated by the portion of the study that was trying to validate the use loss of equilibrium as an earlier endpoint.*

*6. PLOS authors have the option to publish the peer review history of their article (what does this mean?). If published, this will include your full peer review and any attached files.*

Reviewer #1: No

Reviewer #2: No

*While revising your submission, please upload your figure files to the Preflight Analysis and Conversion Engine (PACE) digital diagnostic tool, https://pacev2.apexcovantage.com/. PACE helps ensure that figures meet PLOS requirements. To use PACE, you must first register as a user. Registration is free. Then, login and navigate to the UPLOAD tab, where you will find detailed instructions on how to use the tool. If you encounter any issues or have any questions when using PACE, please email us at figures@plos.org. Please note that Supporting Information files do not need this step.*

---

## [Author Response · Author response to Decision Letter 0]

15 Jan 2020

Please see our point-by-point responses to review comments in the file "Response to reviewers.docx".

---

## [Decision Letter · Decision Letter 1]

26 Feb 2020

PONE-D-19-30081R1

Step-patterned survivorship curves: mortality and loss of equilibrium responses to high temperature and food restriction in juvenile rainbow trout (*Oncorhynchus mykiss*)**

PLOS ONE

Dear Dr. Gosselin,

Thank you for submitting your manuscript to PLOS ONE. After careful consideration, we feel that it has merit but does not fully meet PLOS ONE’s publication criteria as it currently stands. Therefore, we invite you to submit a revised version of the manuscript that addresses the points raised during the review process.

*Please address the minor comments from a re-review of this manuscript by an original reviewer.*

*==============================*

We would appreciate receiving your revised manuscript by Apr 11 2020 11:59PM. To enhance the reproducibility of your results, we recommend that if applicable you deposit your laboratory protocols in protocols.io, where a protocol can be assigned its own identifier (DOI) such that it can be cited independently in the future. For instructions see: http://journals.plos.org/plosone/s/submission-guidelines#loc-laboratory-protocols

*Please include the following items when submitting your revised manuscript:*

*A rebuttal letter that responds to each point raised by the academic editor and reviewer(s). This letter should be uploaded as separate file and labeled 'Response to Reviewers'.**A marked-up copy of your manuscript that highlights changes made to the original version. This file should be uploaded as separate file and labeled 'Revised Manuscript with Track Changes'.**An unmarked version of your revised paper without tracked changes. This file should be uploaded as separate file and labeled 'Manuscript'.*

*Please note while forming your response, if your article is accepted, you may have the opportunity to make the peer review history publicly available. The record will include editor decision letters (with reviews) and your responses to reviewer comments. If eligible, we will contact you to opt in or out.*

*We look forward to receiving your revised manuscript.*

Kind regards,

Madison Powell, PhD

Academic Editor

PLOS ONE

Reviewers' comments:

Reviewer's Responses to Questions

**Comments to the Author**

1. If the authors have adequately addressed your comments raised in a previous round of review and you feel that this manuscript is now acceptable for publication, you may indicate that here to bypass the “Comments to the Author” section, enter your conflict of interest statement in the “Confidential to Editor” section, and submit your "Accept" recommendation.

*Reviewer #1: All comments have been addressed*

*2. Is the manuscript technically sound, and do the data support the conclusions?*

*Reviewer #1: Yes*

*3. Has the statistical analysis been performed appropriately and rigorously? *

*Reviewer #1: Yes*

*4. Have the authors made all data underlying the findings in their manuscript fully available?*

*Reviewer #1: Yes*

*5. Is the manuscript presented in an intelligible fashion and written in standard English?*

*Reviewer #1: Yes*

*6. Review Comments to the Author*

*Reviewer #1: Only some minor comments. The authors needs to check each plot in detail, e.g. unit of mortality rate in Fig 1.C and D. y-axis unit in Suppl Fig S1.*

*7. PLOS authors have the option to publish the peer review history of their article (what does this mean?). If published, this will include your full peer review and any attached files.*

Reviewer #1: No

*While revising your submission, please upload your figure files to the Preflight Analysis and Conversion Engine (PACE) digital diagnostic tool, https://pacev2.apexcovantage.com/. PACE helps ensure that figures meet PLOS requirements. To use PACE, you must first register as a user. Registration is free. Then, login and navigate to the UPLOAD tab, where you will find detailed instructions on how to use the tool. If you encounter any issues or have any questions when using PACE, please email us at figures@plos.org. Please note that Supporting Information files do not need this step.*

---

## [Author Response · Author response to Decision Letter 1]

20 Mar 2020

Please see Response to Reviewers file. Text in that file is copied below:

Comments to the Author

1. If the authors have adequately addressed your comments raised in a previous round of review and you feel that this manuscript is now acceptable for publication, you may indicate that here to bypass the “Comments to the Author” section, enter your conflict of interest statement in the “Confidential to Editor” section, and submit your "Accept" recommendation.

Reviewer #1: All comments have been addressed

Response: Thank you. 

2. Is the manuscript technically sound, and do the data support the conclusions?

Reviewer #1: Yes

Response: Thank you. 

3. Has the statistical analysis been performed appropriately and rigorously? 

Reviewer #1: Yes

Response: Thank you. 

4. Have the authors made all data underlying the findings in their manuscript fully available?

Reviewer #1: Yes

Response: Thank you. 

5. Is the manuscript presented in an intelligible fashion and written in standard English?

Reviewer #1: Yes

Response: Thank you. 

6. Review Comments to the Author

Reviewer #1: Only some minor comments. The authors needs to check each plot in detail, e.g. unit of mortality rate in Fig 1.C and D. y-axis unit in Suppl Fig S1.

Response: Thank you for your careful attention. We have specified the units of the y-axes in Fig 1 C and D, Fig 3, and Supplemental Fig S1. We also provided more clarification for the mortality rate in the Fig 1 caption. 

7. PLOS authors have the option to publish the peer review history of their article (what does this mean?). If published, this will include your full peer review and any attached files.

Do you want your identity to be public for this peer review? For information about this choice, including consent withdrawal, please see our Privacy Policy.

Reviewer #1: No

Response: Thank you for all of your comments and suggested edits. They are much appreciated. 

---

## [Editor Report · Decision Letter 2]

12 May 2020

Step-patterned survivorship curves: mortality and loss of equilibrium responses to high temperature and food restriction in juvenile rainbow trout (*Oncorhynchus mykiss*)**

PONE-D-19-30081R2

Dear Dr. Gosselin,

We are pleased to inform you that your manuscript has been judged scientifically suitable for publication and will be formally accepted for publication once it complies with all outstanding technical requirements.

With kind regards,

Madison Powell, PhD

Academic Editor

PLOS ONE

Additional Editor Comments (optional):

Reviewers' comments:

* *

---

## [Editor Report · Acceptance letter]

14 May 2020

PONE-D-19-30081R2 

Step-patterned survivorship curves: mortality and loss of equilibrium responses to high temperature and food restriction in juvenile rainbow trout (*Oncorhynchus mykiss*) 

Dear Dr. Gosselin:

I am pleased to inform you that your manuscript has been deemed suitable for publication in PLOS ONE. Congratulations! Your manuscript is now with our production department. 

With kind regards,

on behalf of

Dr. Madison Powell 

Academic Editor

PLOS ONE